# Ubiquitin and SUMO conjugation as biomarkers of acute myeloid leukemias response to chemotherapies

Pierre Gâtel[1,2], Frédérique Brockly[1,2], Christelle Reynes[3,4], Manuela Pastore[4], Yosr Hicheri[5], Guillaume Cartron[1,5], Marc Piechaczyk[1,2] (iD), Guillaume Bossis[1,2] (iD)

**Ubiquitin and the ubiquitin-like SUMO are covalently conjugated to thousands of proteins to modulate their function and fate. Many of the enzymes involved in their conjugation are dysregulated in cancers and involved in cancer cell response to therapies. We describe here the identification of biomarkers of the activity of these enzymes and their use to predict acute myeloid leukemias (AML) response to standard chemotherapy (daunorubicin-DNR and cytarabine-Ara-C). We compared the ability of extracts from chemosensitive and chemoresistant AML cells to conjugate ubiquitin or SUMO-1 on 9,000 proteins spotted on protein arrays. We identified 122 proteins whose conjugation by these posttranslational modifiers marks AML resistance to DNR and/or Ara-C. Based on this signature, we defined a statistical score predicting AML patient response to standard chemotherapy. We finally developed a miniaturized assay allowing for easy assessment of modification levels of the selected biomarkers and validated it in patient cell extracts. Thus, our work identifies a new type of ubiquitin-based biomarkers that could be used to predict cancer patient response to treatments.**

## Introduction

Ubiquitin family proteins (collectively called UbL hereafter) are peptidic posttranslational modifiers (Streich & Lima, 2014). The best-characterized ones are ubiquitin and SUMO-1 to -3. SUMO-1 is 50% identical with SUMO-2 and -3, which are 97% identical. UbL are covalently and reversibly conjugated to the lateral chain of lysines from thousands of proteins. Their conjugation involves dedicated enzymatic cascades comprising E1 UbL–activating enzymes (two for ubiquitin, one for SUMO), E2 UbL–conjugating enzymes (46 for ubiquitin, one for SUMO) and several E3 factors (~700 for ubiquitin, ~15 for SUMO) (Streich & Lima, 2014). Ubiquitin can be conjugated to itself via the formation of isopeptide bonds

between its C-terminal glycine and certain of its own lysines (K6, K11, K27, K29, K33, K48, and K63) (Yau & Rape, 2016). SUMO-2 and SUMO-3 can also form chains via SUMOylation of a specific N-terminally located lysine (K11), which is absent in SUMO-1 (Tatham et al, 2001).

Because of the diversity of their target proteins, UbL controls a large range of cellular functions. Like most other posttranslational modifiers, they can either hide or create interaction surfaces on the conjugated protein. The consequences of ubiquitylation also largely depend on the type of chains, K48-linked ubiquitin chains being mostly known to constitute a protein degradation signal recognized by the 26S proteasome (Chau et al, 1989; Glickman & Ciechanover, 2002; Ciechanover, 2017), whereas other types of chains, notably K63- and K11-linked chains, have been involved in protein–protein interactions, signaling, inflammatory response, DNA repair, and ribosomal function (Kwon & Ciechanover, 2017; Haakonsen & Rape, 2019). SUMO is conjugated to more than 6,000, mostly nuclear, proteins. In particular, many proteins involved in gene expression (transcription machinery, transcription factors, transcriptional co-factors, and histones) are regulated upon SUMOylation (Neyret-Kahn et al, 2013; Tempé et al, 2014; Chymkowitch et al, 2015; Rosonina et al, 2017; Cossec et al, 2018). SUMOylation also plays key roles in DNA damage repair via modification of many proteins involved in this process (Garvin & Morris, 2017).

Ubiquitin-like modifiers are critical players in the regulation of numerous cellular pathways and are involved in most, if not all, biological processes. Dysregulation of various enzymes involved in UbL conjugation was found in various cancers with consequences on both tumorigenesis and response to therapies (Mansour, 2018). Among others, these enzymes include E3 ubiquitin ligases such as MDM2 (Carr & Jones, 2016), inhibitor of apoptosis (IAP) (Mohamed et al, 2017), or F-box protein-containing Skp2-cullin-F box (SCF) complexes (Uddin et al, 2016). Overexpression/down-regulation of SUMOylation enzymes has also been reported in many cancers (Seeler & Dejean, 2017), including various hematomalignancies (Boulanger et al, 2019). For instance, the SUMO E2 was shown to be overexpressed in hepatocellular carcinomas, where it participates to resistance to

[1]Institut de Génétique Moléculaire de Montpellier (IGMM), University of Montpellier, CNRS, Montpellier, France   [2]Equipe Labellisée Ligue Contre le Cancer, Paris, France   [3]Institut de Génomique Fonctionnelle (IGF), University of Montpellier, CNRS, INSERM, Montpellier, France   [4]BioCampus Montpellier (BCM), University of Montpellier, CNRS, INSERM, Montpellier, France   [5]Département d'Hématologie Clinique, CHU de Montpellier, Montpellier, France

Correspondence: guillaume.bossis@igmm.cnrs.fr

doxorubicin (Fang et al, 2017) or in multiple myeloma, where it is a marker of bad prognosis (Driscoll et al, 2010). In addition, many components of the SUMO pathway, including E1, E2, and several PIAS E3 proteins were shown to be highly expressed in both B-cell lymphomas and pancreatic cancer overexpressing c-Myc. This results in higher overall SUMO conjugation and consequent vulnerability of these tumors to inhibition of SUMOylation (Hoellein et al, 2014; Biederstädt et al, 2020). Finally, we have recently shown that the SUMO pathway plays an important role, on one side, in the response of acute myeloid leukemias (AML) to standard chemotherapies (Bossis et al, 2014) and, on the other side, in the resistance of non-promyelocytic AML to differentiation therapies using retinoic acid (Baik et al, 2018). UbL, UbL-conjugating/deconjugating enzymes and UbL-conjugated substrates, therefore, constitute a potential new class of biomarkers to predict cancer response to therapies.

AML are a very heterogeneous group of cancers affecting the myeloid lineage. They arise through the acquisition of oncogenic mutations or rearrangements by hematopoietic stem cells or hematopoietic progenitor cells, which, instead of differentiating into normal leukocytes, proliferate and invade the bone marrow. The historical French American British classification of AML was mostly based on the differentiation stage of leukemic cells. It has now been replaced by the World Health Organization classification, which is based on the number and the nature of their genetic abnormalities (Döhner et al, 2017). The latter classification defines groups with favorable, intermediate, or adverse prognosis (Estey, 2012; Dombret & Gardin, 2016). Except for the acute promyelocytic subtype (~10% of AML), which is the only one that can be effectively treated by a differentiation therapy (retinoic acid and arsenic trioxide) (Ng & Chng, 2017), patients diagnosed for all other AML subtypes are generally subjected to a chemotherapeutic treatment that has not significantly changed for the past 40 yr (Dombret & Gardin, 2016). This standard therapy starts with a remission induction treatment combining two genotoxics, one anthracycline (daunorubicin-DNR or idarubicin-IDA) and cytarabine (Ara-C) (3+7 regimen), which is followed by a consolidation treatment using only Ara-C. However, a significant fraction of patients (20–30%) do not respond to the induction treatment and, among those achieving complete remission, relapse rates are high (~40% of 5-yr survival in patients below 60 and ~20% in older ones [Estey, 2012]).

As it is the case for most cancers, no rapidly implementable prognosis tool is currently available at diagnosis to predict the response of AML to standard chemotherapy. This is detrimental for at least two reasons: (i) unnecessary exposure of chemo-resistant patients to severe side effects–generating genotoxics (Minotti et al, 2004) and (ii) loss of time before redirecting refractory patients to novel therapies (targeted therapies, immunotherapies, etc.) and/or enrolling them in clinical trials for innovative therapeutic strategies. Transcriptomic signatures are now used to better stratify patients and adapt treatments in several cancers. However, although various transcriptomic signatures have been defined for AML, none of them is sufficiently reliable to be used in the clinical practice. Proteomic signatures, in particular those based on mass spectrometry, are emerging as promising alternatives (Panis et al, 2016). Even closer to the biological functions dysregulated in cancer, modifomic or PTMomic

analyses, which monitor the activity of enzymes involved in post-translational modifications of proteins, open new perspectives in cancer prognosis and diagnosis (Thygesen et al, 2018).

Alterations in UbL modification being frequently found in AML, we have used this hematomalignancy as a model to address whether global analysis of UbL conjugation could define a new class of biomarkers for patients' response to chemotherapeutic treatments. Using a large-scale protein array–based screening, we identified a UbL signature of AML chemoresistance composed of 122 proteins. We then defined a score based on selected proteins allowing to predict the response to chemotherapy of both AML cell lines and AML patient cells. Finally, we developed a miniaturized assay implementable in clinical routine to rapidly quantify these new biomarkers and validated its prognosis value on a cohort of 37 patients.

## Results

### Analysis of UbL-conjugating activities using protein arrays

To identify potential UbL-modified biomarker proteins, we resorted to protein arrays (Protoarrays; Life Technologies). They display >9,000 recombinant human proteins spotted in duplicate on a nitrocellulose-coated slide. Such arrays have already been used successfully to identify substrates of certain E3 ubiquitin ligases using either total cell extracts (Merbl & Kirschner, 2009) or recombinant enzymes (Gupta et al, 2007; del Rincón et al, 2010). The arrays were incubated with cell extracts either from chemosensitive HL-60 or U937 reference AML cell lines, or from Ara-C-(ARA-R) or DNR-resistant (DNR-R) sublines that we generated from them (Fig 1A). As expected, the chemoresistant sublines showed significantly higher $IC_{50}$ for Ara-C and DNR (Fig 1B). Our assumption was that any dysregulation in the activity of E1, E2, or E3 enzymes in chemoresistant cells would result in changes in UbL substrate modification levels as compared with parental cells. To maximize the reaction efficiency, extracts were complemented with not only recombinant ubiquitin or recombinant-SUMO-1 to avoid rate-limiting amounts of posttranslational modifiers but also vinyl-sulfone–coupled UbLs to block UbL deconjugation via inhibition of UbL-deconjugating enzymes. In each independent experiment, a control condition included a ProtoArray incubated with extracts from parental cells treated with the alkylating agent N-ethylmaleimide (NEM) that inhibits all UbLs E1 and E2 enzymes. Identification of ubiquitin and SUMO-1 substrates and quantification of their modifications were achieved by array scanning after incubation with, first, antibodies directed to either SUMO-1 or the Flag-tag epitope present on exogenous ubiquitin and, then, fluorescent secondary antibodies (Fig 1A). Three independent experiments were performed for all cell lines (24 arrays in total). Signals were corrected for background, normalized using the Protein Array Analyzer (PAA) package (Turewicz et al, 2016), and analyzed using both the Welch and the Wilcoxon–Mann–Whitney statistical tests (see the Materials and Methods section for details). This led to the identification of 988 ubiquitylated and 83 SUMOylated proteins, 72 proteins being both ubiquitylated and SUMOylated (Figs 1C, S1, and Table S1).

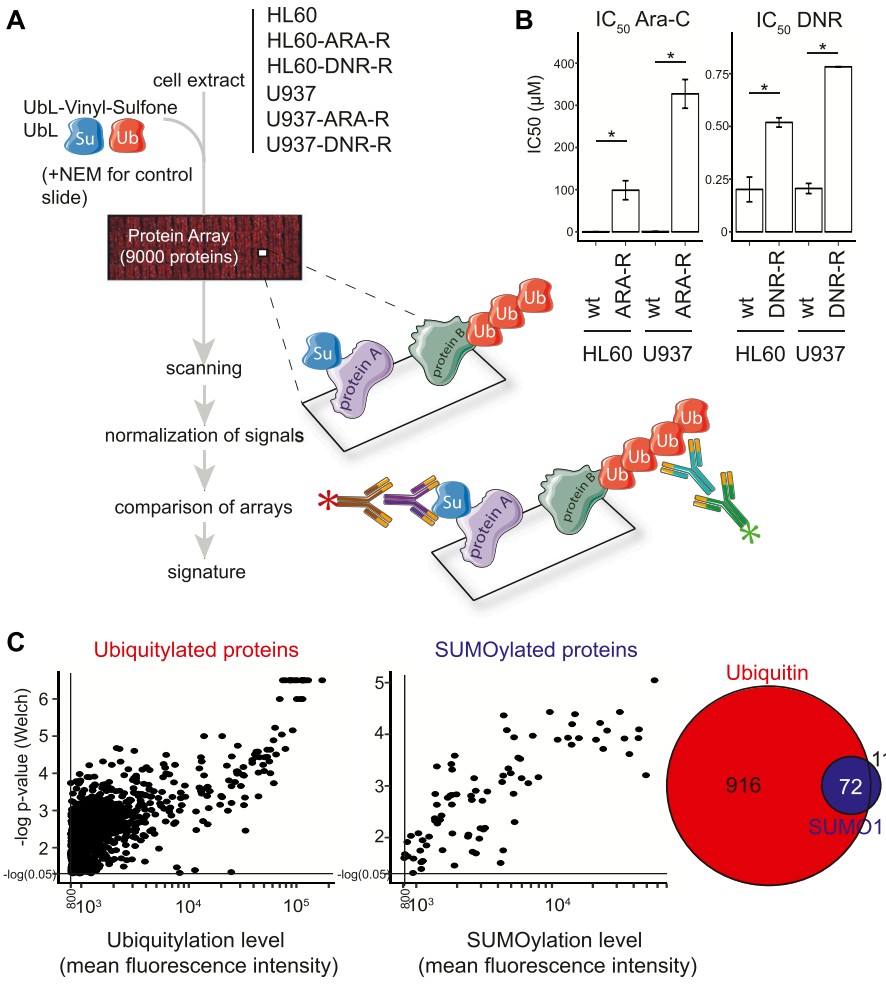

**A**

HL60
HL60-ARA-R
HL60-DNR-R
U937
U937-ARA-R
U937-DNR-R

cell extract

UbL-Vinyl-Sulfone
UbL

(+NEM for control slide)

Protein Array
(9000 proteins)

scanning

normalization of signals

comparison of arrays

signature

**B**

IC$_{50}$ Ara-C    IC$_{50}$ DNR

**C**

Ubiquitylated proteins    SUMOylated proteins

Ubiquitylation level
(mean fluorescence intensity)

SUMOylation level
(mean fluorescence intensity)

Ubiquitin
916    72
SUMO1
11

**Figure 1. Measuring UbL-conjugating activities in acute myeloid leukemias cell extracts using ProtoArrays.**
**(A)** Flowchart followed for UbL signature characterization. Extracts from parental and chemoresistant HL-60 or U937 cells were first supplemented with recombinant UbLs (to avoid rate-limiting amounts of the modifiers) and UbL-vinyl-sulfones (to inhibit UbL deconjugating activities). They were then incubated with protein ProtoArrays. After extensive washes, the arrays were incubated with, first, a primary mouse anti–SUMO-1 antibody and a rabbit anti-Flag antiserum recognizing the Flag-tag present on the recombinant ubiquitin added to the reaction and, then, appropriate fluorescent secondary antibodies. Fluorescence signals were processed using the PAA R package. The statistical analysis was performed to identify a UbL signature of chemoresistance, as described in the Materials and Methods section. Three independent experiments were performed for each cell line. **(B)** IC$_{50}$ of chemosensitive and chemoresistant acute myeloid leukemia cell lines. IC$_{50}$ of chemosensitive parental HL-60 and U937 (wt) cells and of their resistant counterparts (see the Materials and Methods section) (ARA-R and DNR-R) was assayed after 24 h of exposure to drugs. n = 3, Mean ± SEM, paired *t* test with * corresponding to *P* < 0.05. **(C)** Identification of ubiquitylated- and SUMOylated proteins. Normalized Ub and SUMO-1 fluorescence data obtained on all arrays incubated with extracts were compared with averaged signals on control arrays (extracts supplemented with NEM to inhibit UbL conjugation activities) to identify robustly UbL-conjugated proteins. Proteins showing significant differences between the two groups when using both the Welch- and the Wilcoxon–Mann–Whitney statistical tests and having mean fluorescence intensity values higher than 800 (arbitrary threshold) on ProtoArrays were selected for further analysis. The Venn diagram shows the number or proteins identified as modified by SUMO-1 and/or ubiquitin.

## Identification of a UbL signature of AML resistance to standard chemotherapy

Among the proteins identified as robustly ubiquitylated or SUMOylated on the arrays, we then selected those that were differentially modified between cell extracts from chemoresistant and parental cell lines. A first analysis was performed by comparing the pooled data obtained for all resistant cell lines to those from all sensitive ones. This led to the identification of 52 proteins differentially modified by ubiquitin and 27 proteins differentially modified by SUMO-1 (Figs 2A, S2, S3, and Table S2), four of them being differentially modified by both UbLs. To identify biomarkers that might specify specific AML subtypes or resistance to one of the two drugs, we also performed a second analysis in which the data for each cell line (HL-60 and U937) and each resistance (DNR and Ara-C) were considered separately. Although the sample size for each condition was smaller than in the first analysis, and consequently the statistical significances lower, we identified 65 proteins for ubiquitin and 12 for SUMO-1 that were differentially modified in at least one of the resistant cell lines, as compared with its parental counterpart (Fig 2B and Table S3). Among the 65 ubiquitylated proteins, 42 had not been identified in the global analysis. For SUMO-1–modified proteins, eight new substrates were identified in

the second analysis. Altogether, the compilation of all ProtoArray data (global and separated analyses) identified a signature of AML chemoresistance comprising 94 ubiquitylated and 35 SUMOylated proteins, making a total of 122 individual proteins (Table S4). An ontology analysis showed that these are principally involved in the ubiquitin-proteasome pathway, stress response, DNA damage repair, and autophagy, which are all processes often dysregulated in chemoresistant cancer cells (Fig 2C).

## Generation of a UbL score to predict patients' response to chemotherapeutic drugs

To further validate the signature, in particular on patient samples, we selected 23 of the 122 proteins that showed both a high level of modification and the most robust signal differences between sensitive and resistant cell lines in the global or separated analyses (Table S4). In a first step, we used them to generate a UbL score aiming at predicting patients' response to chemotherapy. To this aim, we applied a genetic algorithm (GA) (Scrucca, 2013) to select groups of proteins pertinent for predicting chemotherapy resistance in cell lines and in patients. We used 30 variables corresponding to the 23 selected proteins (7 SUMOylated + 9 ubiquitylated + 2 × 7 SUMOylated and ubiquitylated) (see the

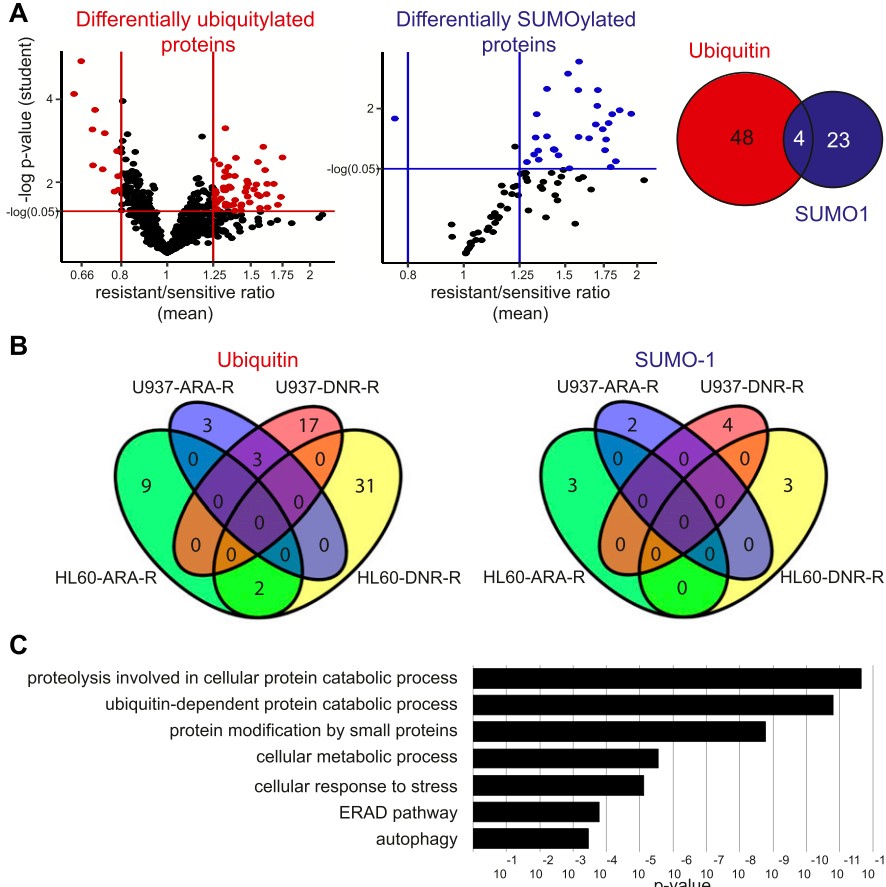

**Figure 2. UbL–conjugated protein signature of chemoresistance.**
**(A)** Identification of UbL-modified biomarkers of acute myeloid leukemias chemoresistance. Modification levels of the proteins modified by ubiquitin (left panel) or SUMO-1 (central panel) selected in Fig 1C were compared between all parental (U937 and HL-60) and drug-resistant (ARA-R or DNR-R) sublines. Differentially modified proteins with significant *P*-values in both Wilcoxon signed rank- and one sample *t* test and with a drug-resistant versus parental cell ratio higher than 1.25 or lower than 0.8 are indicated in red for ubiquitylated proteins and in blue for SUMOylated ones. The Venn diagram shows the overlap between differentially ubiquitylated- and SUMOylated proteins. **(B)** Identification of UbL-conjugated biomarkers specific for HL-60 and U937 cell resistance to Ara-C or DNR. Statistical analyses between drug-resistant and parental cells were performed separately for U937 and HL-60 cell lines and for each drug resistance. The number of proteins showing a significant *P*-value in one sample *t* test and a ratio between drug-resistant and parental cells higher than 1.5, or lower than 0.66, are shown. **(C)** Ontology analysis of the UbL signature. An ontology analysis of the 122 proteins of the UbL signature was performed using the Panther software.

Materials and Methods section). The GA was run on the cell line datasets multiple times, with different parameter settings to obtain a total of 40 solutions (combinations of selected proteins). Of the 30 variables, 25 were selected at least once in the 40 solutions obtained. They were ranked by their frequency of selection (Fig 3A), as frequently selected proteins are more likely to be predictive. We then divided this list in five embedded subsets containing an increasing number of selected variables/proteins (Fig 3A) for linear discriminant analysis (LDA), which uses an optimized linear combination of all variables to assign observations to target classes (here, sensitive or resistant). As expected, we observed that the most frequently selected proteins were present in the most predictive selected solutions, which suggested that they were not random artefacts because of the GA.

To validate the approach, we first used the LDA on cell lines with all the subsets and found that the solution containing the top 7 proteins in the GA (subset 2) was the most efficient and could predict whether the cell line was sensitive or resistant to DNR or Ara-C in 16 cases of 18 tested (six cell lines and three replicates) (Fig 3B). To determine if it could also be used to predict AML patients' response to these drugs, we resorted to bone marrow aspirates from four patients, all of them being of the rather immature M1 subtype in the French American British classification. Two of these patients were sensitive to induction chemotherapy (anthracycline + cytarabine) and two were refractory (Table S5). These samples were used to prepare extracts and to probe ProtoArrays as described above for cell lines. Interestingly, the LDA containing "subset 2" proteins could predict the response to chemotherapy of all of the four patients tested (Fig 3C). Although a higher number of patient samples still need to be tested for stronger validation of the approach, our data suggested that the biomarkers we identified might be used to predict AML patients' response to chemotherapies.

## Development of a miniaturized assay to measure signature protein modification by UbLs

Protoarrays are useful to identify signatures. However, they are difficult to implement for routine clinical diagnosis because of the number of cells required, the challenge of their standardization, and their high cost. To circumvent these difficulties, we developed a miniaturized flow cytometry assay to further evaluate the prognosis value of the biomarkers we identified. This assay is based on the use of Luminex xMap beads, which are color-coded magnetic beads. To validate it, we cloned the 23 selected proteins (Table S4) as fusions with GST. 10 of them could be produced in *Escherichia coli* and were coupled to differently colored xMap beads. The coupled beads were then multiplexed and incubated with cell extracts. They were then analyzed by flow cytometry using, first, an anti–SUMO-1 or anti-Flag tag antibody to quantify modification by SUMO-1 and ubiquitin, respectively, and, then,

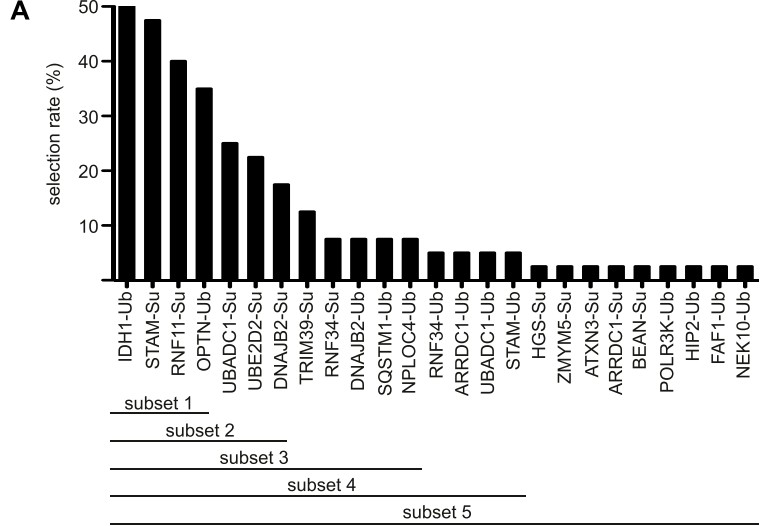

**Figure 3. Generation of a UbL-predictive score of acute myeloid leukemias response to chemotherapies.**

**(A)** Selection of the best predictive proteins. A list of 30 predictors among the 122 signature proteins was chosen to run a genetic algorithm. These predictors corresponded to 23 proteins modified either by ubiquitin (Ub), SUMO (Su), or both UbLs. The rate of selection of each of these proteins and its associated UbL (Ub or Su) is indicated on the graph. The proteins were separated in five subsets for linear discriminant analysis analysis. **(B, C)** Probability of acute myeloid leukemias sensitivity/resistance using the subset 2 solution. **(B, C)** The seven proteins showing the highest rate of selection in the genetic algorithm (subset 2) were used for linear discriminant analysis to predict the probability of resistance for the U937 and HL60 cell lines (B) or patient samples (C). Cells were considered sensitive to chemotherapy if the probability to belong to the group of resistant cells was below 50% and resistant if over 50%.

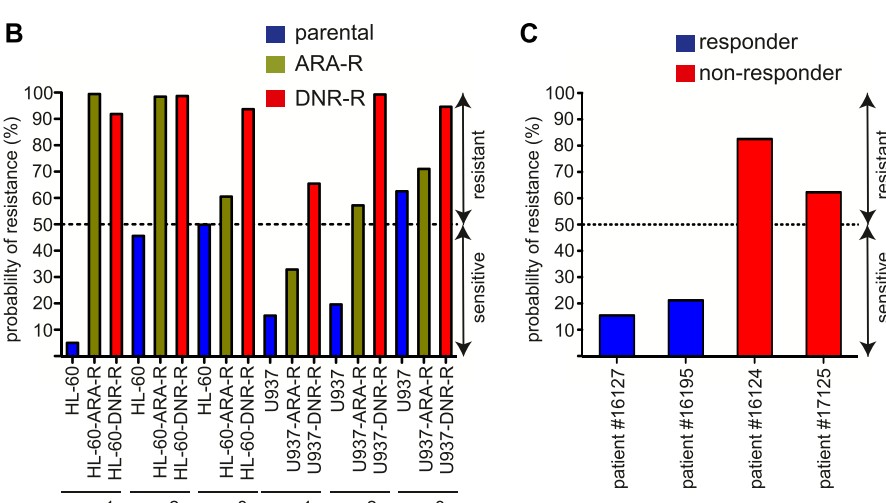

specific secondary antibodies linked to two different fluoro-chromes (Fig 4A). If ubiquitylation was detected on all proteins tested, this was, however, not the case for SUMOylation. A most likely explanation for this discrepancy is that ubiquitin can make long polyubiquitin chains, facilitating their detection on protein substrates, whereas SUMO cannot. We consequently focused the rest of our study on ubiquitylated proteins. Of the 10 proteins tested, four proteins (UBADC1, STAM, SQSTM1, and HIP2) showed significant differences in their ubiquitylation levels between parental, Ara-C– and/or DNR-resistant U937 (Fig 4B and C). Interestingly, although all proteins were incubated simultaneously in the same extracts, differences could be detected in their modification. For example, the highest ubiq-uitylation of STAM was obtained using extracts from Ara-C-resistant cells, whereas UBADC1 and HIP2 showed increased modification only with DNR-resistant cell extracts. Altogether, these data suggested that bead-based detection of ubiq-uitylation provides an easy and quantitative way to detect specific alterations in the ubiquitylation cascade and identify

STAM, UBADC1, SQSTM1, and HIP2 as potential Ub-modifiable biomarkers of AML chemoresistance.

### Analysis of UbL signature proteins in primary AML cells using the miniaturized flow cytometry assay

To further validate the use of the bead-based assay, we used cell extracts prepared from bone marrow aspirates from 37 patients who responded, or not, to induction chemotherapy. We found that cell extracts obtained from patients who did not respond to in-duction chemotherapy resulted in higher levels of in vitro ubiq-uitylation of STAM, UBADC1, or SQSTM1 compared with those from patients who responded (Fig 4D and Table S5). 6 of 10 refractory patients showed high ubiquitylation levels for at least one of the biomarkers. In most cases, not all the biomarkers had high ubiq-uitylation levels, suggesting that the combination of biomarkers has higher predictive value than each biomarker alone. Altogether, this suggested that the ubiquitylation signature proteins, comprising

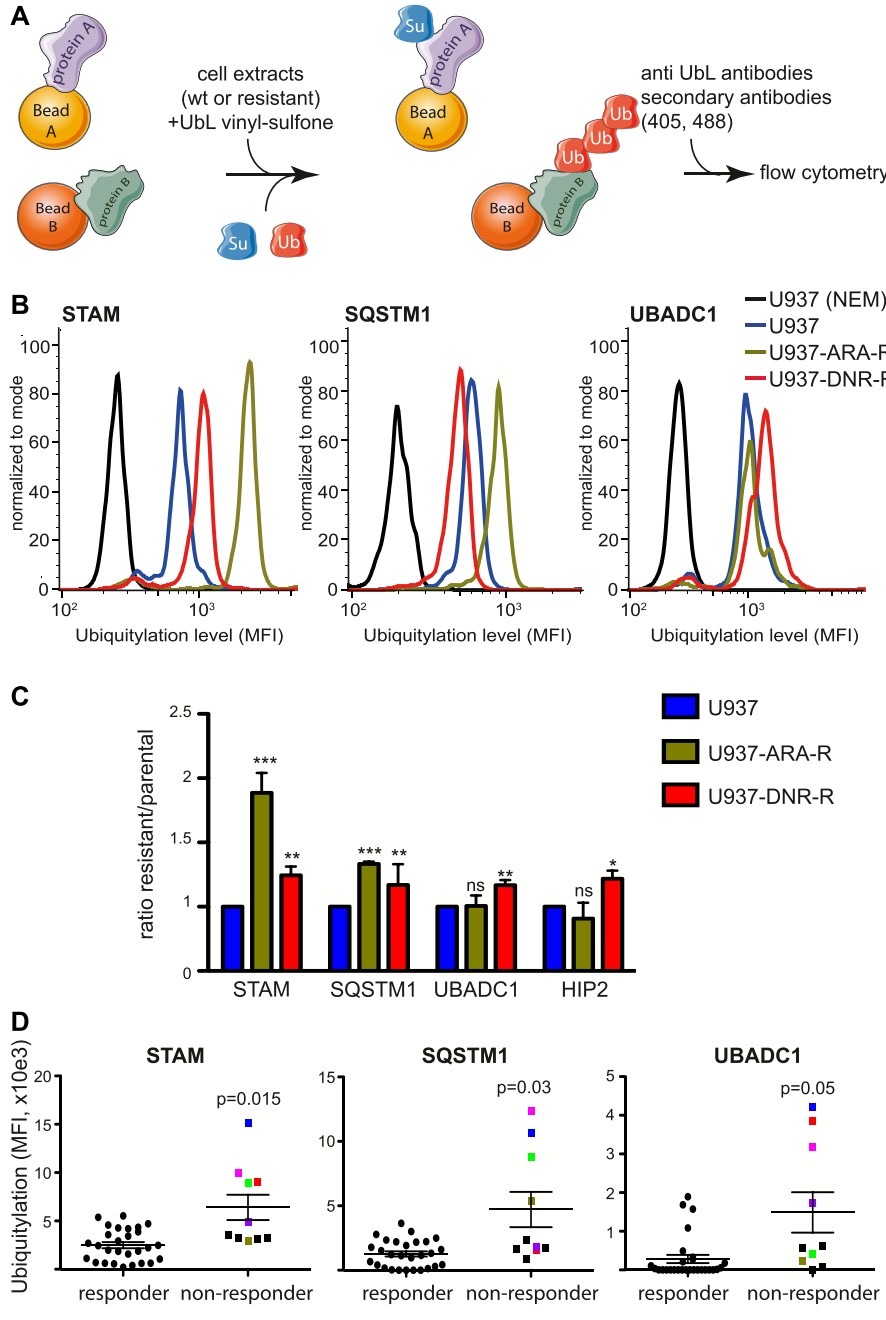

**Figure 4. Flow cytometry assay for the detection of UbL-conjugated chemoresistance biomarkers in cell lines and patient samples.**
**(A)** Principle of the assay. Recombinant signature proteins are produced in *E. coli* and coupled to differently colored xMap beads. Coupled beads are then mixed and incubated with cell extracts supplemented with recombinant UbL, in the presence of UbL-vinyl-sulfones to inhibit deconjugating activities. Protein modification is quantified by flow cytometry using a combination of primary antibodies directed to UbLs and fluorescent secondary antibodies. **(A, B, C)** Analysis of STAM, SQSTM1, UBADC1, and HIP2 ubiquitylation. Extracts from parental, ARA-R, or DNR-R U937 cells were used as described in (A) with beads coupled to STAM, SQSTM1, UBADC1 and HIP2. **(B)** Representative flow cytometry profiles for their ubiquitylation are shown in (B). **(C)** Quantification are presented in (C). For quantification, background (cell extracts supplemented with NEM to inhibit UbL conjugation activities) was subtracted and ratio between resistant and parental cell lines was shown (n = 6 for STAM, n = 3 for SQSTM1, UBADC1 and HIP2). Mean ± SEM. Paired *t* test, *$P < 0.05$, **$P < 0.01$, ***$P < 0.001$. **(D)** Extracts from bone marrow aspirates from 37 patients at diagnosis were used in a multiplexed flow cytometry analysis using xMap beads coupled to STAM, UBADC1, and SQSTM1. 29 patients responded to induction chemotherapy (<10% of blasts in bone marrow 30 d after the beginning of chemotherapy) and 10 did not (>10% of blasts in bone marrow 30 d after the beginning of chemotherapy). Mean ± SEM. Unpaired *t* test with Welch's correction. For the refractory patients, those showing high ubiquitylation for at least one of the biomarkers are colored-coded. MFI, median fluorescence intensity; ns, not significant.

among them STAM, UBADC1, and SQSTM1, might serve as a therapeutic response biomarker in AML.

## Discussion

In summary, our work points to a new class of biomarkers based on ubiquitin and SUMO conjugation that might be used to predict the response of AML patients to standard chemotherapies. In addition, we have developed a miniaturized assay that allows the quantification of these biomarkers using low amounts of cells and amenable to the use of patient samples.

Dysregulations of the ubiquitin- and SUMO-conjugated proteome in cancer cells can be linked to increased/decreased expression of their respective E1/E2/E3 conjugating enzymes, which could be (partly) monitored using transcriptomic approaches. However, in most cases, they are due to changes in their activities, which is more challenging to monitor, in particular in patient samples. Mass spectrometry has been widely used to analyze, at the proteome scale, cell protein SUMOylome and ubiquitylome (Hendriks & Vertegaal, 2016; Heap et al, 2017). However, although the sensitivity of this technology increases rapidly, it still requires large amounts of material, which is generally not compatible with the use of patient samples as starting material. Moreover, the biological

lability of the isopeptide bond linking UbLs to their substrates renders their mass-spectrometry identification particularly challenging. We used here protein arrays to monitor variations in the activity of UbL-conjugating enzymes. The protein array technology is endowed with several advantages. First, as it relies on enzymatic activities, it is amenable to relatively small numbers of cells as compared with mass spectrometry approaches. Second, it does not rely on the abundance of the spotted proteins because they are present in excess and in similar amounts on the arrays. Finally, it allows the analysis of two different UbLs in the same sample/experiment. We focused our study on ubiquitin and SUMO conjugation, as these modifiers are the best-characterized and best-studied UbLs. We identified 988 ubiquitylated and 83 SUMOylated proteins that are robustly modified on the protein arrays using cell extracts. These proteins can, thus, be used as biomarkers of the activity of the enzymes (E1, E2, and E3s) responsible for their ubiquitin and/or SUMO modification in vitro. Of note, this assay does not monitor the activity of UbL-deconjugating enzymes because they are inhibited with UbL-vinyl-sulfones. Many of the conjugated proteins we identified are themselves involved in UbL conjugation/deconjugation pathways. This probably reflects their higher ability to interact with the enzymes of the pathway, including via non-covalent interactions with ubiquitin or SUMO. It is, however, important to underline that the UbL-target proteins we identified are not necessarily genuine targets of the concerned UbL-conjugating enzymes in vivo. At least one explanation for this is that many enzymes of the UbL cycle, as well as substrate proteins, are spatially partitioned in living cells and may never (or little) meet in vivo, a situation which is no longer preserved when resorting to cell extracts in an in vitro assay.

We show here that alterations of ubiquitin and SUMO conjugation pathways are associated with AML resistance to DNR and Ara-C. We identified 122 proteins, the ubiquitylation/SUMOylation of which changes when using cell extracts from AML cell lines resistant to DNR or Ara-C comparatively to cell extracts from their chemotherapy-sensitive counterparts. In most cases, their ubiquitylation/SUMOylation is increased, suggesting an over-activation of specific UbL enzymes in chemoresistant AML cells. Only a small fraction (around 10%) of the ubiquitin/SUMO-modifiable proteins, we identified on the arrays are differentially modified between chemosensitive and chemoresistant cells. This suggests that, not the entire Ub/SUMO pathways, but only specific enzymes within these pathways are dysregulated in chemoresistant cells. These are most probably E3 enzymes because dysregulation of E1 or E2s would alter a much larger repertoire of substrates. Analysis of the differentially modified proteins we discovered did not allow to identify the dysregulated E3 or specific drug resistance pathways known to be altered in chemoresistant AML. Identifying, which are the dysregulated E3 enzymes could pave the way to the development of new therapeutic strategies to improve AML treatment.

Among the 122 UbL signature proteins, some are differentially modified only in DNR- or in Ara-C–resistant cells or only in one of the two model AML cell lines. This suggests that some of the alterations resulting in changes in UbL enzyme activities are specific to the genotoxics they are resistant to and/or to the AML subtype. We then generated scores based on the most robust proteins of the signature in terms of level and differences of UbL modification between the parental and the resistant AML cell lines we used as models. These scores were used on protein array data obtained with four patients' samples and one of the scores could retrospectively predict the response of these four patients to the chemotherapy. Even if the number of patients' samples was too small to validate this assay on a definitive statistical basis, our work nevertheless provides a proof of concept that such an approach could be used to help predict AML patients' response to current standard chemotherapies.

To further validate the biomarkers we identified, in particular in patients' samples, it was important to develop a miniaturized assay that would require less cells and be easily and rapidly (few hours) implementable for future use in clinical practice. The assay is based on the Luminex technology, which uses multiplexable color-coded microbeads. This technology is routinely used, including in clinical analysis services, to detect circulating hormones, chemokines, cytokines, or autoantigens using beads coupled with antibodies specific for these proteins. Here, instead of antibodies, we coupled purified recombinant proteins from the UbL-signature to the beads. The modified beads (one protein per bead color) were then multiplexed and incubated with cellular extracts, which were prepared under conditions preserving UbL-conjugating enzymes activities. This assay could precisely quantify the level of coupled protein modification by ubiquitin and, thereby, precisely estimate the differences in their modification levels between samples. Among the proteins that we analyzed, we identified four proteins, STAM, UBADC1, SQSTM1, and HIP2, which are differentially ubiquitylated with extracts from cell lines that were resistant to chemotherapies compared with cells that are sensitive. It should, however, be noted that not all proteins identified on the ProtoArrays could be validated in the xMap bead–based assay. At this stage of investigations, we do not exclude that this might be due to differences in their conformation and/or differences in preexisting posttranslational modifications, as they were produced in insect cells in the former case and in bacteria in the latter one. Along the same line, improvement of protein production and detection methods could also allow to better detect SUMOylation of the proteins we identified using the protein arrays and improve the prognosis efficiency of the assay. Importantly, we could validate the predictive potential of the ubiquitylation of three of the biomarkers (STAM, UBADC1, and SQSTM1) on a cohort of 37 patients. Their ubiquitylation was significantly higher in refractory patients compared with those who responded to chemotherapies. 6 of 10 refractory patients showed high ubiquitylation of at least one of the biomarkers. Future development of this assays will require to analyze more biomarkers of the signature to lower the number of false negative (4 of 10).

AML is a therapeutic emergency, as patients most often need to start chemotherapy in the next few days after diagnosis. It is, however, critical to rapidly determine the risk/benefit ratio before starting current standard chemotherapy to which severe side effects are associated. Decision between intensive standard chemotherapy, epigenetic therapies (e.g., azacitidine), and best-supportive care is currently mostly based on the age, fitness, and co-morbidities of the patient. Cytogenetics, in particular the number of genetic abnormalities, can also be used as a prognosis marker. However, it generally takes 1 or 2 wk for the clinicians to get

the results. Most of the time, the treatment needs to be started before having these results. We have established the basis of an assay, based on UbL modifications by patients' cell extracts, which could be performed at diagnosis in a few hours together with flow cytometry analyses routinely carried out to confirm AML diagnosis. Its results, combined with the abovementioned criteria, might therefore help clinicians to propose the best therapeutic option to each patient, including redirection towards novel treatments or ongoing clinical trials, notably for emerging drugs targeting ubiquitin or SUMO pathways.

# Materials and Methods

### Cell culture

HL-60 and U937 cells (DSMZ) were cultured at 37°C in Roswell Park Memorial Institute medium supplemented with 10% FBS and streptomycin/penicillin in the presence of 5% $CO_2$. Both cell lines were authenticated by the American Type Culture Collection using short tandem repeat analysis. All cells were regularly tested negative for mycoplasma. After thawing, the cells were passaged at a density of $3 \times 10^5$/ml every 2–3 d for not more than 10 passages. U937 and HL-60 cells resistant to Ara-C (cytarabine) and DNR were generated by culture in the presence of increasing drug concentrations (up to 0.1 $\mu$M for Ara-C and 0.03 $\mu$M for DNR) for 2–3 mo.

### Patient samples

Bone marrow aspirates were collected after obtaining written informed consent from patients under the frame of the declaration of Helsinki and after approval by the institutional "Sud Méditerranée 1" Ethical Committee (ref 2013-A00260-45; HemoDiag collection). Fresh leukocytes were purified by density-based centrifugation using Histopaque 1077 (Sigma-Aldrich) and used immediately for extract preparation. Detailed characteristics and treatments of the patients involved in this study are provided in Table S5.

### $IC_{50}$ measurement

Cells were seeded at a concentration of $3 \times 10^5$/ml in Roswell Park Memorial Institute medium complemented with 0.1, 1, 10, 50, 100, or 250 $\mu$M of Ara-C (Sigma-Aldrich) or 0.01, 0.05, 0.1, 0.5, 1, or 10 $\mu$M of daunorubicin (Sigma-Aldrich). Viability was measured 24 h later using the MTS assay from Promega following the manufacturer's protocol. $IC_{50}$ was calculated using the GraphPad PRISM software.

### Cellular extracts

Cell lines grown at a $5–8 \times 10^5$/ml density or patient's bone marrow aspirates were spun down (300$g$) at 4°C for 5 min and washed once with PBS. After pellet resuspension in 1 ml of PBS, they were centrifuged again (16,000$g$) at 4°C for 5 min. Pellets were resuspended and incubated at 4°C for 30 min in a hypotonic buffer (Hepes, pH 7.5, 20 mM, $MgCl_2$ 1.5 mM, KCl 5 mM, DTT 1 mM, and 1 mg/l of aprotinine, leupeptin, and pepstatin) in a volume of either (i) 100 $\mu$l per $50 \times 10^6$ cells to generate concentrated extracts used for ProtoArray probing or (ii) 25 $\mu$l per $2 \times 10^6$ cells in the case of extracts used in xMap bead–based flow cytometry assays. Cell lysis was achieved through four freezing/thawing cycles using liquid nitrogen, and DNA was sheared owing to 10 passages through a 20-1/2G needle. Extracts were finally centrifuged twice (16,000$g$) at 4°C for 20 min and supernatants were aliquoted, flash-frozen, and kept at –80°C until use.

### Protein arrays

Human protein arrays (ProtoArrays V5.0; Life Technologies) kept at –20°C were equilibrated at 4°C for 15 min and, then, saturated with the ProtoArray Blocking Buffer (PA055; Life Technologies) supplemented with Synthetic Block (PA017; Life Technologies) and 1 mM DTT at 4°C for 1 h. To probe ProtoArrays for UbL protein modification, cell extracts were first supplemented with 5 $\mu$M ubiquitin-vinyl sulfone, 2.5 $\mu$M SUMO1-vinyl sulfone, and 2.5 $\mu$M SUMO2-vinyl sulfone (Boston Biochem) to inhibit the corresponding UbL-deconjugating enzymes. Control extracts were also incubated with 50 mM NEM (Sigma-Aldrich) to inhibit any UbL conjugation. The extracts were then supplemented with Tween-20 (0.1%), ATP (1 mM), and home-made recombinant SUMO-1 or SUMO-2 (15 $\mu$M, produced as previously described [Bossis et al, 2005]), or Flag-ubiquitin (30 $\mu$M; Boston Biochem) and were immediately laid on ProtoArray slides, which were then covered with a coverslip and incubated at 30°C for 1 h in a humidified atmosphere. Arrays were washed three times for 5 min with a washing buffer (PBS, pH 7.4, Tween 20 0.1%, and 1× Synthetic Block) supplemented with 0.5% SDS and, then, twice for 5 min with only the washing buffer. Next, they were incubated in the washing buffer under gentle agitation at 4°C for 1 h with 1 $\mu$g/ml of rabbit anti-Flag (F7425; Sigma-Aldrich) or mouse anti–SUMO-1 (21C7 from the Developmental Studies Hybridoma Bank) antibodies. After five washes of 5 min in the washing buffer, they were incubated at 4°C for 90 min with 0.5 $\mu$g/ml of Alexa Fluor 647–labelled antimouse- or Alexa Fluor 546–labelled antirabbit antibodies (Thermo Fisher Scientific). Arrays were washed five times for 5 min with the washing buffer, once with $H_2O$, and finally, dried by centrifugation before fluorescence scanning using the Innoscan 710 device from Innopsys.

### Analysis of ProtoArray data

Measured fluorescence intensities were associated to the corresponding protein ID according to their coordinates on the arrays using the Mapix software (Innopsys). Intensities were processed using the PAA R package (Turewicz et al, 2016). In brief, (i) duplicated protein spot intensities were averaged using the LoadGPR function and (ii) the background was corrected using the BackgroundCorrect function. Fluorescence intensities within the same experiment were normalized by quantiles using the NormalizeArray function. Finally, intensities were normalized between the different experiments using the BatchAdjust function.

### Data filtering

To identify the proteins modified by ubiquitin or SUMO-1 on the ProtoArrays, we first had to filter proteins displaying signal

significantly higher in UbL conjugation-permissive conditions comparatively to non-permissive ones (i.e., control conditions using NEM-treated extracts). The classical *t* test was not adapted to exploit our results, as variances could be very different between UbL conjugation-permissive and non-permissive conditions. We, therefore, selected proteins with significant *P*-values (lower than 0.05) in both the parametrical Welch- and the non-parametrical Wilcoxon–Mann–Whitney tests. Then, proteins having an averaged normalized fluorescence intensity value less than 800 (arbitrary threshold) were filtered out to obtain a list of robustly modified proteins.

### Identification of differentially UbL-modified proteins between chemosensitive and chemoresistant cells

The signal intensity ratios between the parental and the drug-resistant cells for the proteins selected after data filtering (see above) were calculated using both a Wilcoxon signed-rank test and a one sample *t* test. This analysis was first performed on all arrays (parental, DNR-, or ARA-c–resistant U937 and HL60 cell lines). In this case, the sample size being large, we considered as differentially modified all proteins showing *P*-values < 0.05 in both tests. For the analysis on the individual cell lines (U937 and HL-60) and drugs (Ara-C and DNR), the sample size being smaller, we considered as differentially modified the proteins showing *P*-values < 0.05 in the one sample *t* test.

### Ontology analysis

Ontology analyses were performed using the Panther software.

### GA and linear discriminant analysis

The R package GA, providing a GA (Scrucca, 2013), was used and its code is provided as Supplemental Data 1. The aim was to determine the right number of variables to create a parsimonious predictive model. GAs are mathematical models inspired by Charles Darwin's model of natural selection. The natural selection preserves only the fittest individuals over the different generations. An evolutionary algorithm improves the selection over time and allows the best solution to emerge from the best of prior solutions. The selected features were then tested with LDA using the R package MASS (Ripley, 1996; Venables & Ripley, 2002). This mathematical method uses a linear combination of all variables to assign observations to target classes. To this aim, it creates a decision rule based on the n available variables (score = $\alpha \times R_1 + \beta \times R_2 + \ldots + \omega \times R_n$) by maximizing the between-class variability and minimizing the within-class one. A cross-validation step that separates the observations in two groups (training dataset on which the model is established and test dataset on which the model is validated) was performed to get more robust results.

### Production of recombinant proteins

cDNAs encoding the proteins of interest were recovered from the Ultimate ORF library (Thermo Fisher Scientific) and cloned in the bacterial expression vector pGGWA (Busso et al, 2005) vector using the Gateway technology according to the manufacturer's protocol (Life Technologies). Constructs were then transformed in the BL21 (DE3) *E. coli* strain. Protein production was induced with 1 mM isopropyl $\beta$-D-1-thiogalactopyranoside (IPTG) for 6 h in bacteria exponentially growing at 25°C. Bacterial pellets were resuspended in Tris–HCl, pH 8.6, 50 mM containing NaCl 500 mM and MgSO$_4$ 50 mM and flash-frozen in liquid N$_2$. After thawing, bacterial suspensions were supplemented with 1 mg/ml lysozyme (Sigma-Aldrich), 8 mM $\beta$-mercaptoethanol, and 1 mg/l of aprotinin, leupeptin and pepstatin and incubated at 4°C for 1 h. Bacterial debris were spun down at 100,000$g$ for 1 h. The extract was then bound to glutathione agarose beads (Generon) equilibrated in buffer A (Tris–HCl, pH 8.6, 50 mM, NaCl 150 mM, MgSO$_4$ 50 mM, $\beta$-mercaptoethanol 8 mM, and 1 $\mu$g/l of aprotinin, leupeptin and pepstatin). The column was then extensively washed with buffer A and eluted by addition of 20 mM reduced glutathione (Sigma-Aldrich).

### Protein coupling to xMap

$2 \times 10^5$ magnetic xMap beads (low concentration) from Luminex were transferred to a low-binding microtube (Eppendorf) and washed using NaCl 500 mM. They were then resuspended in 50 $\mu$l of MES (2-ethanesulfonic acid), pH 6.1, 50 mM and incubated in the presence of 5 $\mu$g/ml 1-ethyl-3-(3-dimethylaminopropyl)carbodiimide hydrochloride (EDC; Pierce) and 5 $\mu$g/ml Sulfo-NHS (Pierce) at room temperature for 20 min under agitation. Beads were then washed in PBS containing 500 mM NaCl and incubated with 7 $\mu$g of recombinant protein to be coupled in 100 $\mu$l PBS at room temperature for 2 h. They were then washed twice with PBS containing 0.1% BSA, 0.02% Tween 20, 0.05% sodium azide, and 500 mM NaCl and stored at 4°C in PBS containing 0.1% BSA, 0.02% Tween-20, and 0.05% sodium azide.

### UbLs conjugation to proteins coupled to xMap beads

SUMO-1-, SUMO-2-, and ubiquitin-vinyl sulfones (0.5 $\mu$M each) were added to diluted cellular extracts (10 $\mu$l), which were incubated at 4°C for 15 min. Control extracts were also incubated with 100 mM NEM. We then added to the extract 10$^3$ protein-coupled xMap beads contained in 10 $\mu$l of a reaction buffer containing 20 mM Hepes, pH 7.3; 110 mM KOAc; 2 mM Mg(OAc)$_2$; 0.05% Tween-20; 0.5 mM EGTA; 0.2 mg/ml ovalbumine; 1 mM DTT; 1 mg/l aprotinin, leupeptin, and pepstatin; 1 mM ATP; 30 $\mu$M Flag-ubiquitin; 15 $\mu$M SUMO-1; and 15 $\mu$M SUMO-2. Reactions were performed at 30°C for 45 min. Beads were washed twice for 5 min with PBS containing 0.05% Tween-20 and 0.5% SDS and three times for 5 min with PBS containing 0.05% Tween-20. They were then incubated with 1 $\mu$g/ml of mouse anti-SUMO-1 (21C7) and rabbit anti-Flag antibodies for 1 h under agitation at room temperature. After washing in PBS containing 0.05% Tween-20 for 5 min, they were incubated for 30 min at room temperature with antimouse Alexa Fluor 488 and antirabbit Alexa Fluor 405 antibodies in 100 $\mu$l of PBS containing 0.05% Tween 20. Beads were again washed for 5 min with PBS containing 0.05% Tween 20. They were then resuspended in 200 $\mu$l PBS and flow cytometry analyzed using the LSR Fortessa device from BD Biosciences. Results were analyzed using the FlowJow software.

## Supplementary Information

## Acknowledgements

We are grateful to the Institut de Génétique Moléculaire de Montpellier (IGMM) "Oncogenesis and Immunotherapy" group members and Dr O Coux for fruitful discussions and critical reading of the manuscript. We thank the Montpellier Genomic Collection and Montpellier Ressources Imagerie platforms for technical assistance. Funding was provided by the Centre National de la Recherche Scientifique (CNRS), Ligue Nationale contre le Cancer (Programme Equipe Labellisée), Région Languedoc-Roussillon ("Chercheur d'Avenir" contract), Association Laurette Fugain (ALF-2017/02 contract), the Fédération Leucémie Espoir, and the ANR ("Investissements d'avenir" program; ANR-16-IDEX-0006). P Gâtel was supported by fellowships from the University of Montpellier, the Montpellier Centre Hospitalo Universitaire, and the Fondation Association pour la Recherche contre le Cancer (ARC). The HEMODIAG_2020 collection of clinical data and patient samples was funded by the Montpellier University Hospital, the Montpellier SIRIC, and the Languedoc-Roussillon Region.

### Authors Contribution

P Gâtel: conceptualization, formal analysis, validation, investigation, and methodology.
F Brockly: validation, investigation, and methodology.
C Reynes: conceptualization, formal analysis, and methodology.
M Pastore: data curation.
Y Hicheri: resources.
G Cartron: resources.
M Piechaczyk: conceptualization.
G Bossis: conceptualization.

### Conflict of Interest Statement

The authors declare that they have no conflict of interest.

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
