## [Reviewer comments · Life Science Alliance]

Life Science Alliance

Ubiquitin and SUMO conjugation as biomarkers of Acute Myeloid Leukemias response to chemotherapies

Pierre Gâtel, Frédérique Brockly, Christelle Reynes, Manuela Pastore, Yosr Hicheri, Guillaume Cartron, Marc Piechaczyk, and Guillaume Bossis

DOI: <https://doi.org/10.26508/lsa.201900577>

Corresponding author(s): Guillaume Bossis, CNRS/ Montpellier University

Review Timeline:

Submission Date:	2019-10-11
Editorial Decision:	2019-11-19
Appeal Request:	2020-02-28
Editorial Decision:	2019-02-28
Editorial Decision:	2020-04-01
Revision Received:	2020-04-06
Accepted:	2020-04-06

Scientific Editor: Andrea Leibfried

Transaction Report:

November 19, 2019

Re: Life Science Alliance manuscript #LSA-2019-00577

Dr. Guillaume Bossis
CNRS/ Montpellier University
Institut de Génétique Moléculaire de Montpellier
1919 route de Mende
Montpellier, I 34293
France

Dear Dr. Bossis,

Thank you for submitting your manuscript entitled "Ubiquitin and SUMO conjugation as biomarkers of Acute Myeloid Leukemias response to chemotherapies" to Life Science Alliance. The manuscript has now been seen by expert reviewers, whose reports are appended below.

As you will see, your manuscript received somewhat split views from the reviewers. Reviewer #1 points out that the work is too preliminary at this stage and that follow up studies will be complicated given the dis-continued protein arrays. Reviewer #2 and #3 are more positive, but note that the predictive power of the approach and thus the potential clinical value is limited. We have discussed your work further within our editorial team in light of these reports. I am afraid we came to the conclusion that the value to others is too limited to allow further consideration here. We are thus returning your manuscript to you with the message that we cannot publish it in Life Science Alliance.

We are sorry our decision is not more positive, but hope that you find the reviews constructive. Of course, this decision does not imply any lack of interest in your work and we look forward to future submissions from your lab.

Thank you for your interest in Life Science Alliance.

Sincerely,

Reviewer #1 (Comments to the Authors (Required)):

Summary: AML is a deadly disease that is normally treated with chemotherapy. However, a proportion of AML patients do not receive benefit from chemotherapy. Therefore, there is a unmet clinical need to identify which patients will likely benefit from chemotherapy or would be better suited for treatment with targeted or experimental therapy. This manuscript uses a protein microarray-based approach to identify ubiquitin and SUMO biomarkers of AML response to chemotherapy. The study compares ubiquitin and SUMO conjugation activities for AML cell lines and specimens that are either sensitive or resistant to chemotherapy to identify potential biomarkers of response. The authors identify a Ub/SUMO activity signature that has predictive value in determining responders and developed a selected flow cytometry-based biomarker assay and validated it using AML patient extracts. The manuscript is well written, data well presented, and a description of the translational potential of identifying a Ub/SUMO activity signature that could be used to predict AML response to chemotherapy is clearly articulated.

Strengths: The authors generated chemotherapy resistant AML cell lines and compared the ubiquitin and SUMO conjugation profiles with that of parental cells for 2 different chemotherapy treatments. This identified 122 differentially ubiquitylated/SUMOylated substrates on the protein array, with 23 proteins showing robust modification and signal differences. The 23 substrate-based activity profile could predict 16/18 cases of the resistant/non-resistant cell lines and 2 resistant and 2 sensitive AML specimens. The authors then express 3 of these proteins in bacteria and develop a flow cytometry-based assay to detect their ubiquitylation activity using AML cell lines and patient specimens, which showed some predictive value in determining responsive cells/specimens. The flow cytometry-based analysis is innovative and potentially clinically useful since protein array-based analyses are cost prohibited.

Weaknesses: The study is primarily a proof of concept that protein arrays could be used to generate a Ub/SUMO activity profile to predict treatment response. However, the study falls short of validating a Ub/SUMO activity signature that can actually predict chemotherapy response as only 2 responders vs. non-responders are tested in follow-up experiments using the protein arrays. In addition, follow up studies by this group or others to test this concept will be complicated by the fact that the protein arrays used in this study are no longer commercially available. The authors then attempt to develop a flow cytometry-based methodology to profile Ub activity in AML specimens, but only 3/10 proteins expressed in bacteria. The authors are correct in pointing out that insect cell expression was used in the development of the protein array to maintain some PTMs and it is unclear why insect cells were not used for protein expressions in these experiments. Ubiquitylation of the 3 proteins using AML cell lines and AML patient specimens does not convincingly show their ubiquitylation associates with response to chemotherapy- the cell line results are mixed and only 5 AML non-responder specimens are analyzed making it difficult to draw any conclusions. Overall, the study is pre-mature and the discovery and translation of a Ub/SUMO biomarker signature that has power in predicting AML response to chemotherapy will require a more robust analysis of AML patient specimens and further development of the flow cytometry assay.

Reviewer #2 (Comments to the Authors (Required)):

This manuscript uses two quite novel approaches to identify the ubiquitin and SUMO modifications in cells. The authors propose to identify the modifications by ubiquitin and SUMO which distinguish the differential response of Acute Myeloid Leukemia cell lines to standard chemotherapy. The way the study is set up is overall good and some of the findings have been extended to patient samples with encouraging results. I value highly the fact that the study is using very novel approaches to translate findings on the ubiquitin system in the clinical practice

However, I have some concerns outlined below which would require addressing before publication :

1. The authors keep referring to a "modifomic" signature. Although the term is appealing, I am not sure it helps the reader in comprehending the significance of the findings.
2. Figure 1C could highlight some of the modified proteins identified in the graphs reported. Is any of the proteins identified been related previously to resistance to ARA-C and DNR? I think it would be nice to emphasise the protein differentially identified on microarrays which is the main discovery of the manuscript
3. Is there any statistical test which could validate that the comparison of the signals in arrays provides good information?
4. Figure 2A and B I would have preferred to see some hits highlighted in the graphs or a table summarising the relevant hits
5. Figure 2C the ontology analysis reveals simply that there is an enrichment for ubiquitylation process which is ok but wouldn't the authors expect something else?
6. Figure 3 It was unclear to me how the authors are matching their genes subsets to patient samples.
7. Figure 3 B and C the statistic of replicates is lacking
8. Figure 4 the assay is quite clever and nice and most probably modifying it with E3 ligases could give very good results but currently I am afraid that the predictive value of this test is quite limited. The robustness of the approach for influencing clinical decision should be much higher.
9. The authors could have commented more on the biology of the hits identified and their relationship to AML and drug resistance.

Reviewer #3 (Comments to the Authors (Required)):

The manuscript by Gatel et al. utilizes protein arrays to discover biomarkers to predict response of AML cells to chemotherapy. By screening for both ubiquitin and SUMO modified proteins, the authors identified a UbL-conjugated proteins signature of chemoresistance and generated a score to predict AML response to chemotherapy. Furthermore, they generated a flow-cytometry based assay for the detection of UbL conjugates biomarkers that were further tested in AML cell lines and patients.

Overall, this is very interesting and well done study which deserves to be published as soon as possible.

I have two recommendations:

- 1) Can the authors try K48 and K11 specific antibodies to assess whether chain specific linkage further increases association to AML chemoresistance in cell lines and patients.
- 2) Can the authors increase the number of primary AML patients to strengthen the prediction

correlation?

Dear Editor,

We thank you for having considered the manuscript by Gatel et al. "Ubiquitin and SUMO conjugation as biomarkers of Acute Myeloid Leukemia response to chemotherapies" and for having provided us with a constructive feedback on our work (Editorial Decision LSA-2019-00577, dated November 19, 2019).

Two reviewers (Reviewers 2 and 3) were strongly supportive (though asking for some clarifications) and one (Reviewer 1), -although acknowledging the innovative nature and the clinical significance of our work-, concluded it to be too preliminary for publication, his/her main point being that a "more robust analysis of AML patient specimens" would be necessary.

To address Reviewer 1's major point and to improve the significance of our work, we have extended our study via analyzing a much larger cohort of Acute Myeloid Leukemia patients (37 patients). This allowed us to statistically validate that the *in vitro* level of ubiquitylation of the biomarkers we identified actually constitutes a marker of AML response to chemotherapies. In addition to this major point, we have also addressed all other points raised by the three referees.

A point-by-point response to all Reviewers' comments is to be found below. Modifications were borne to the main text and to certain figures to take these comments into account and to include our new data. Three Supplementary Figures were also added to the manuscript.

We are fully aware that you receive far more manuscript than you can publish. However, due to the serious improvement of our work and the fact that we have met all Reviewers' points, including Reviewer 1 major comment, we hope that you would be willing to consider the revised version of our manuscript. Our work can pave the way to the development of new prognosis tools, which are, unfortunately, still critically lacking for the treatment of AML patients. In addition, the approach and the tools we developed will be of great use for both basic and translational sciences, as well as to better analyze and understand ubiquitin-like proteins pathways and their dysregulations. For all of these reasons, we feel that the Gatel et al.'s work should be of high interest for the readership of Life Science Alliance.

Sincerely yours,
Guillaume Bossis, for the authors

Point by point response to referees

We thank all reviewers for their insightful comments. We have addressed all their comments and modified the manuscript accordingly.

- **Reviewer 1**

Summary: AML is a deadly disease that is normally treated with chemotherapy. However, a proportion of AML patients do not receive benefit from chemotherapy. Therefore, there is a unmet clinical need to identify which patients will likely benefit from chemotherapy or would be better suited for treatment with targeted or experimental therapy. This manuscript uses a protein microarray-based approach to identify ubiquitin and SUMO biomarkers of AML response to chemotherapy. The study compares ubiquitin and SUMO conjugation activities for AML cell lines and specimens that are either sensitive or resistant to chemotherapy to identify

potential biomarkers of response. The authors identify a Ub/SUMO activity signature that has predictive value in determining responders and developed a selected flow cytometry-based biomarker assay and validated it using AML patient extracts. The manuscript is well written, data well presented, and a description of the translational potential of identifying a Ub/SUMO activity signature that could be used to predict AML response to chemotherapy is clearly articulated.

Strengths: The authors generated chemotherapy resistant AML cell lines and compared the ubiquitin and SUMO conjugation profiles with that of parental cells for 2 different chemotherapy treatments. This identified 122 differentially ubiquitylated/SUMOylated substrates on the protein array, with 23 proteins showing robust modification and signal differences. The 23 substrate-based activity profile could predict 16/18 cases of the resistant/non-resistant cell lines and 2 resistant and 2 sensitive AML specimens. The authors then express 3 of these proteins in bacteria and develop a flow cytometry-based assay to detect their ubiquitylation activity using AML cell lines and patient specimens, which showed some predictive value in determining responsive cells/specimens. The flow cytometry-based analysis is innovative and potentially clinically useful since protein array-based analyses are cost prohibited.

Weaknesses:

Weakness 1 raised by Reviewer 1: *The study is primarily a proof of concept that protein arrays could be used to generate a Ub/SUMO activity profile to predict treatment response. However, the study falls short of validating a Ub/SUMO activity signature that can actually predict chemotherapy response as only 2 responders vs. non-responders are tested in follow-up experiments using the protein arrays. In addition, follow up studies by this group or others to test this concept will be complicated by the fact that the protein arrays used in this study are no longer commercially available.*

Reply to Weakness 1: Reviewer 1 is correct that the Protoarrays we have been using have been discontinued. This indeed explain why we could not validate the score we created on a larger number of patients. However, protein arrays are an emerging technology and other companies are providing such whole proteome arrays (<https://cambridgeproteinarrays.com/HuProt.php>) and other companies are developing custom-made arrays (<https://www.raybiotech.com/protein-array/#protein-arrays>). Therefore, follow-up studies will be possible. However, Protoarrays are not suitable for the screening of large patient cohorts, as they require a large number of cells (50 millions), which can only be obtained for few hyper-leucocytic patients. This explains why we have developed the miniaturized, flow cytometry-based assay to monitor the biomarkers we identified with the Protoarrays using patients' material. We have now used this assay on a cohort of 37 patients and validate, for 3 of the biomarkers, that their level of ubiquitylation is generally higher in patients that are refractory to chemotherapies (see below).

Weakness 2 raised by Reviewer 1: *The authors then attempt to develop a flow cytometry-based methodology to profile Ub activity in AML specimens, but only 3/10 proteins expressed in bacteria. The authors are correct in pointing out that insect cell expression was used in the development of the protein array to maintain some PTMs and it is unclear why insect cells were not used for protein expressions in these experiments.*

Reply to Weakness 2: As pointed by all 3 reviewers, this assay is innovative and could be used in clinical practice. Reviewer 1 expressed concerns that only 3/10 proteins were purified. There was actually a misunderstanding, as we could express and purify 10/23 proteins from the signature in bacteria. Out of these 10 proteins, 4 showed significant differences between

chemosensitive and chemoresistant U937 cell line. Since this number was sufficient to provide a proof of concept of the relevance of the assay on patient samples, we did not resort to insect cell production, which requires specific equipment and know-how we do not have. However, as stated in the manuscript, insect cells could be used to further develop the prognosis assay.

Weakness 3 raised by Reviewer 1: Ubiquitylation of the 3 proteins using AML cell lines and AML patient specimens does not convincingly show their ubiquitylation associates with response to chemotherapy- the cell line results are mixed and only 5 AML non-responder specimens are analyzed making it difficult to draw any conclusions.

Reply to Weakness 3: To address this comment, we have performed a new retroactive experiment on frozen samples from 37 patients, 10 of them being non-responders. This allowed us to obtain, for the 3 most robust biomarkers (UBADC1, STAM and SQSTM1), statistically significant higher ubiquitylation of these proteins using extracts from patients that are refractory to the chemotherapy compared to those who are responders. Six out of the 10 refractory patients showed high ubiquitylation of at least one of the biomarkers. These results are now presented in Figure 4D.

- **Reviewer 2**

This manuscript uses two quite novel approaches to identify the ubiquitin and SUMO modifications in cells. The authors propose to identify the modifications by ubiquitin and SUMO which distinguish the differential response of Acute Myeloid Leukemia cell lines to standard chemotherapy. The way the study is set up is overall good and some of the findings have been extended to patient samples with encouraging results. I value highly the fact that the study is using very novel approaches to translate findings on the ubiquitin system in the clinical practice

However, I have some concerns outlined below which would require addressing before publication

Point 1 raised by Reviewer 2: The authors keep referring to a "modifomic" signature. Although the term is appealing, I am not sure it helps the reader in comprehending the significance of the findings.

Reply to Point 1: We agree and have removed this term from our manuscript

Point 2 raised by Reviewer 2: Figure 1C could highlight some of the modified proteins identified in the graphs reported. Is any of the proteins identified been related previously to resistance to ARA-C and DNR? I think it would be nice to emphasize the protein differentially identified on microarrays which is the main discovery of the manuscript

Reply to Point 2: We have now added 3 supplementary figures providing the names of the most significantly modified or differentially modified proteins. We did not find known correlations between the differentially modified proteins and resistance to Ara-C or DNR. This is due to the fact that the proteins we identified are markers of Ub/SUMO enzymatic activities and not necessarily genuine in vivo targets.

Point 3 raised by Reviewer 2: Is there any statistical test which could validate that the comparison of the signals in arrays provides good information?

Reply to Point 3: The work presented in this manuscript has largely involved statisticians for all steps of the analyses. In particular, to compare all arrays and get the list of all robustly modified proteins, we used both the parametrical Welch- and the non-parametrical Wilcoxon-

Mann-Whitney (WMW) tests. We then used Wilcoxon signed-rank test and a one sample *t*-test to identify those, which are differentially modified. The detail of these analysis is described in the Methods part of the manuscript.

Point 4 raised by Reviewer 2: *Figure 2A and B I would have preferred to see some hits highlighted in the graphs or a table summarising the relevant hits*

Reply to point 4: We have now added two supplementary Figure (Sup Fig 2 and Sup Fig 3), where we have highlighted relevant hits.

Point 5 raised by Reviewer 2: *Figure 2C the ontology analysis reveals simply that there is an enrichment for ubiquitylation process which is ok but wouldn't the authors expect something else?*

Reply to point 5: The biomarkers we identified are not necessarily endogenous targets of the dysregulated pathways involved in AML resistance to chemotherapies. They are biomarkers of UbL enzymatic activities, which are themselves dysregulated. The relevance of the identified biomarkers is now better discussed.

Point 6 raised by Reviewer 2: *Figure 3 It was unclear to me how the authors are matching their genes subsets to patient samples.*

Reply to Point 6: Protoarray experiments were performed for the 4 patients tested. For each of the subsets 2 biomarkers (7 proteins) selected in the genetic algorithms (Figure 3A), a LDA (linear discriminant analysis) was performed to assign the observation (ubiquitylation level of the biomarker on the protoarray probed with patient cell extract) to a specific group (sensitive or resistant).

Point 7 raised by Reviewer 2: *Figure 3B and C the statistic of replicates is lacking*

Reply to Point 7: For Figure 3B, we have chosen to show individual replicates rather than the mean of all replicates to provide a better view of the results. The differences between the parental and resistant groups are however significant in most cases (HL60 parental vs ARA-R $p=0.05$, HL60 parental vs DNR-R $p=0.01$, U937 parental vs DNR-R $p=0.04$). For patient samples (Figure 3C), each patient sample could be used only on one Protein Array. No replicate could be performed since we could only get enough cells for one array. This limitation for the use of Protoarrays with patient samples explains why we developed the flow cytometry-based assay, which was now validated on a cohort of 37 patients (see below).

Point 8 raised by Reviewer 2: *Figure 4 the assay is quite clever and nice and most probably modifying it with E3 ligases could give very good results but currently I am afraid that the predictive value of this test is quite limited. The robustness of the approach for influencing clinical decision should be much higher.*

Reply to Point 8: We have now used the assay using a cohort of 37 patients (instead of 17 initially). This allowed us to statistically validate its predictive potential. The new results are presented in Figure 4D. We did not add E3 ligases to our assay since it relies on the use of enzymatic machinery present in the extract. Our hypothesis is that dysregulation of Ubiquitin or SUMO conjugation enzymes are a marker of AML resistance to chemotherapy.

Point 9 raised by Reviewer 2: *The authors could have commented more on the biology of the hits identified and their relationship to AML and drug resistance.*

Reply to Point 9: As stated above, we did not find any relevant link between the biomarkers and AML response to chemotherapies. This is likely due to the fact that these proteins are

markers of dysregulated enzymatic activities and not endogenous targets of these enzymes. This point is now better discussed in the manuscript.

- **Reviewer 3**

The manuscript by Gatel et al. utilizes protein arrays to discover biomarkers to predict response of AML cells to chemotherapy. By screening for both ubiquitin and SUMO modified proteins, the authors identified a Ubl-conjugated proteins signature of chemoresistance and generated a score to predict AML response to chemotherapy. Furthermore, they generated a flow-cytometry based assay for the detection of Ubl conjugates biomarkers that were further tested in AML cell lines and patients. Overall, this is very interesting and well done study which deserves to be published as soon as possible.

Point 1 raised by Reviewer 3: *Can the authors try K48 and K11 specific antibodies to assess whether chain specific linkage further increases association to AML chemoresistance in cell lines and patients.*

Reply to Point 1: We agree with Reviewer 3 that ubiquitin-specific chains could increase the complexity and therefore the prognosis value of the signature. This would however require to identify K48- or K11-linked substrate using the Protoarrays, which is beyond the scope of the present work.

Point 2 raised by Reviewer 3: *Can the authors increase the number of primary AML patients to strengthen the prediction correlation?*

Reply to Point 2: We have now used the flow cytometry-based assay in a retrospective experiment with a cohort of 37 patients (instead of 17 initially). This allowed us to statistically validate its predictive potential. The new results are presented in Figure 4D.

MS: LSA-2019-00577

Dr. Guillaume Bossis
CNRS/ Montpellier University
Institut de Génétique Moléculaire de Montpellier
1919 route de Mende
Montpellier, I 34293
France

Dear Dr. Bossis,

Your manuscript entitled "Ubiquitin and SUMO conjugation as biomarkers of Acute Myeloid Leukemias response to chemotherapies" has now been reconsidered, and I am pleased to let you know that we have decided to send your manuscript for external re-review.

We will let you know when the reviews have been received and a decision has been made.

Yours sincerely,

April 1, 2020

RE: Life Science Alliance Manuscript #LSA-2019-00577R-A

Dr. Guillaume Bossis
CNRS/ Montpellier University
Institut de Génétique Moléculaire de Montpellier
1919 route de Mende
Montpellier, I 34293
France

Dear Dr. Bossis,

Thank you for submitting your revised manuscript entitled "Ubiquitin and SUMO conjugation as biomarkers of Acute Myeloid Leukemias response to chemotherapies". As you will see, reviewer #1 appreciates the introduced changes and we would thus be happy to publish your paper in Life Science Alliance pending final revisions necessary to meet our formatting guidelines:

- Please upload your manuscript file in docx format
- Please upload all figures, including supplementary figures, as individual files
- Please remove the panel descriptor (A) from the legends of figures S2 and S3
- Please mention the statistical test used next to the mentioned p-values in the figure legend 1B
- I think it would be good to include the algorithm used (GA), unless it was not altered in any way

A. FINAL FILES:

- An editable version of the final text (.DOC or .DOCX) is needed for copyediting (no PDFs).
- High-resolution figure, supplementary figure and video files uploaded as individual files: See our detailed guidelines for preparing your production-ready images, <http://www.life-science-alliance.org/authors>
- Summary blurb (enter in submission system): A short text summarizing in a single sentence the

study (max. 200 characters including spaces). This text is used in conjunction with the titles of papers, hence should be informative and complementary to the title. It should describe the context and significance of the findings for a general readership; it should be written in the present tense and refer to the work in the third person. Author names should not be mentioned.

B. MANUSCRIPT ORGANIZATION AND FORMATTING:

Thank you for your attention to these final processing requirements.

Sincerely,

Reviewer #1 (Comments to the Authors (Required)):

Gatel et al. have done an excellent job addressing the concerns raised in the previous review. The authors now provide additional tables for the reader and have expanded their testing of the biomarkers to 37 AML patients with known response to chemotherapy. The paper provides proof of principle that a robust set of Ub/SUMO biomarkers could ultimately be identified that predict chemotherapy response in AML. They also lay the groundwork for a flow cytometry-based biomarker assay that circumvents many of the problems of adapting a protein microarray-based assay to clinical applications and will allow for a rapid prediction of patient response. By addressing these concerns, the manuscript is now much stronger and warrants publication.

April 6, 2020

RE: Life Science Alliance Manuscript #LSA-2019-00577RR

Dr. Guillaume Bossis
CNRS/ Montpellier University
Institut de Génétique Moléculaire de Montpellier
1919 route de Mende
Montpellier, I 34293
France

Dear Dr. Bossis,

Thank you for submitting your Research Article entitled "Ubiquitin and SUMO conjugation as biomarkers of Acute Myeloid Leukemias response to chemotherapies". It is a pleasure to let you know that your manuscript is now accepted for publication in Life Science Alliance. Congratulations on this interesting work. Please provide us with the algorithm as soon as possible to allow a smooth production process.

DISTRIBUTION OF MATERIALS:

Again, congratulations on a very nice paper. I hope you found the review process to be constructive and are pleased with how the manuscript was handled editorially. We look forward to future exciting

submissions from your lab.

Sincerely,
